# Deep sequencing of *Escherichia coli* exposes colonisation diversity and impact of antibiotics in Punjab, Pakistan

Tamim Khawaja [1,2,3,13], Tommi Mäklin [4,13], Teemu Kallonen [5,13], Rebecca A. Gladstone [6], Anna K. Pöntinen [6,7], Sointu Mero[2,3], Harry A. Thorpe [6], Ørjan Samuelsen [7,8], Julian Parkhill [9], Mateen Izhar[10], M. Waheed Akhtar[11], Jukka Corander [4,6,12] ✉ & Anu Kantele [1,2,3] ✉

Multi-drug resistant (MDR) *E. coli* constitute a major public health burden globally, reaching the highest prevalence in the global south yet frequently flowing with travellers to other regions. However, our comprehension of the entire genetic diversity of *E. coli* colonising local populations remains limited. We quantified this diversity, its associated antimicrobial resistance (AMR), and assessed the impact of antibiotic use by recruiting 494 outpatients and 423 community dwellers in the Punjab province, Pakistan. Rectal swab and stool samples were cultured on CLED agar and DNA extracted from plate sweeps was sequenced en masse to capture both the genetic and AMR diversity of *E. coli*. We assembled 5,247 *E. coli* genomes from 1,411 samples, displaying marked genetic diversity in gut colonisation. Compared with high income countries, the Punjabi population generally showed a markedly different distribution of genetic lineages and AMR determinants, while use of antibiotics elevated the prevalence of well-known globally circulating MDR clinical strains. These findings implicate that longitudinal multi-regional genomics-based surveillance of both colonisation and infections is a prerequisite for developing mechanistic understanding of the interplay between ecology and evolution in the maintenance and dissemination of (MDR) *E. coli*.

*Escherichia coli* is a leading cause of mild and severe infections worldwide, and the frequent emergence of novel virulent and resistant pandemic clones represents a severe clinical concern. During the past decade, extensive genomic surveillance of *E. coli*, encompassing strains from both colonisation and disease, has deepened our understanding of factors contributing to virulence, shaped the map of resistance, and uncovered trends in population prevalence of clones across high-income countries (HICs)[1–6]. Several whole-

[1]Meilahti Infectious Diseases and Vaccine Research Center (MeiVac), Helsinki University Hospital and University of Helsinki, Helsinki, Finland. [2]Human Microbiome Research Program, University of Helsinki, Helsinki, Finland. [3]Multidiciplinary Center of Excellence in Antimicrobial Resistance Research, FIMAR, Medical Faculty, University of Helsinki, Helsinki, Finland. [4]Department of Mathematics and Statistics, University of Helsinki, Helsinki, Finland. [5]Department of Clinical Microbiology, Turku University Hospital, Turku, Finland. [6]Department of Biostatistics, University of Oslo, Oslo, Norway. [7]Norwegian National Advisory Unit on Detection of Antimicrobial Resistance, Department of Microbiology and Infection Control, University Hospital of North Norway, Tromsø, Norway. [8]Department of Pharmacy, Faculty of Health Sciences, UiT The Arctic University of Norway, Tromsø, Norway. [9]Department of Veterinary Medicine, University of Cambridge, Cambridge, UK. [10]Department of Microbiology, Shaikh Zayed Post-Graduate Medical Institute, Lahore, Pakistan. [11]School of Biological Science, University of the Punjab, Lahore, Pakistan. [12]Parasites and Microbes, Wellcome Sanger Institute, Hinxton, UK. [13]These authors contributed equally: Tamim Khawaja, Tommi Mäklin, Teemu Kallonen. ✉e-mail: jukka.corander@medisin.uio.no; anu.kantele@helsinki.fi

genome sequencing (WGS)-based studies of Extended-Spectrum β-Lactamase (ESBL)-producing *E. coli* genetic variation have been conducted in Asia[7–10], identifying a notable difference in the diversity of genetic lineages compared with Northern Europe and the US. In addition to narrowing the focus on ESBL-producing *E. coli*, none of these studies have used sweeps of bacterial growth from the enrichment medium followed by DNA extraction and whole-genome sequencing en masse to capture a more complete picture of the genetic diversity of *E. coli* in their samples. Thus, understanding remains limited about both the wider population diversity of *E. coli* lineages in Asia and the genetic diversity of *E. coli* colonising hosts at any single point in time, either with or without the influence of recent antibiotic use.

Recently, using Norwegian and British parallel surveillance data spanning nearly two decades of *E. coli* bacteremia surveillance, Pöntinen et al. demonstrated that the extent of non-penicillin β-lactam antibiotic use is a likely driver of the prevalence of the virulent ESBL-producing ST131 C2 clone in low-to-moderate antibiotic usage populations[5]. Furthermore, they showed that resistance frequencies and outpatient antibiotic usage generally remained de-coupled across different antibiotic classes and genetic lineages in such high-income settings. How these drivers are reflected in the distributions of resistant and susceptible lineages in regions with poor hygiene and extensive use of antibiotics remains unknown, which motivated us to conduct the current study.

Our study drawing upon WGS data presents the first opportunity to not only quantify the within-host genetic diversity of *E. coli* at strain-level resolution, but also assess the impact of antibiotic use on this diversity. Moreover, this investigation estimates the population distribution of *E. coli* genetic lineages in Punjab, Pakistan, and makes comparisons with the distributions estimated from genomic surveillance data collected in Norway, UK and the U.S. We demonstrate that multiple clones of *E. coli* circulating in Punjab are shared internationally, as members of these clones can be identified among carbapenemase-producing isolates from patients in Europe. Our data

further suggest that some of the endemically circulating and clinically important clones in Europe sporadically turn up in Pakistan but do not become endemic in this region.

## Results

### *Escherichia coli* carriage in the Punjab province of Pakistan

A total of 994 patients provided rectal swab samples at the Manga Mandi outpatient clinic (Fig. 1a). Of these, 497 delivered follow-up samples, one was excluded for missing information on antibiotic use, and two for a possible mixup of samples. The final cohort thus included 494 patients, out of which 258 (52.2%) were women/girls. Most patients were adults (433, 87.7%), with a median age of 38 years (IQR 28–50). Community samples (Fig. 1a) were provided by 423 volunteers, 208 (49.2%) of whom were women/girls. The median age of the volunteers was 28 years (IQR 12–42). There were 140 (33.1%) children, 13 of whom were two years old or younger.

We analysed the short-read sequencing data obtained from the samples using the mSWEEP/mGEMS tools[11,12], which together form a high-resolution computational metagenomics pipeline capable of resolving the composition of a sequencing data sample at a user-defined taxonomic level (see the Methods section for details). We resolved the samples with a hierarchical approach (Fig. 1b), first performing an initial species screening step on the samples as described in a previous study[6] to exclude reads not originating from *E. coli*, and secondly resolving the composition of the *E. coli* reads to the level of PopPUNK sequence clusters. These clusters were defined to roughly correspond to *E. coli* clonal complexes, which are groups of similar sequence types (ST) previously defined on the basis of an allele typing scheme. Our approach resulted in 5247 high-quality (as defined in the 'Methods' section) bin-assembled genomes (BAGs). All results are reported for the high-quality BAGs unless noted otherwise and we refer to the BAGs assembled from bins corresponding to the same PopPUNK cluster with the name of the most common *E. coli* sequence type contained in the cluster.

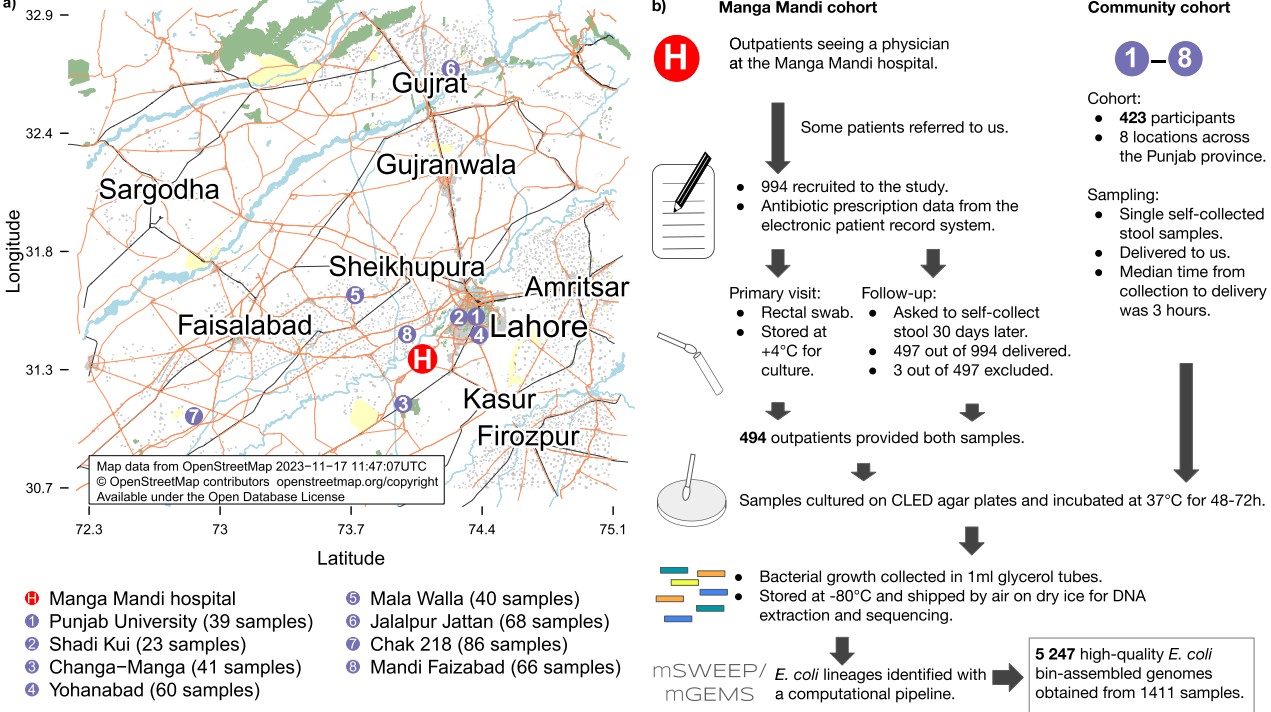

**Fig. 1 | Description of the sampling.** Panel (**a**) shows the location of the Manga Mandi hospital near Lahore (red circle with a white "H") and the community sampling locations (purple numbered circles). Panel (**b**) describes the cohort recruitment and sampling procedure. Source data are provided as a Source Data file.

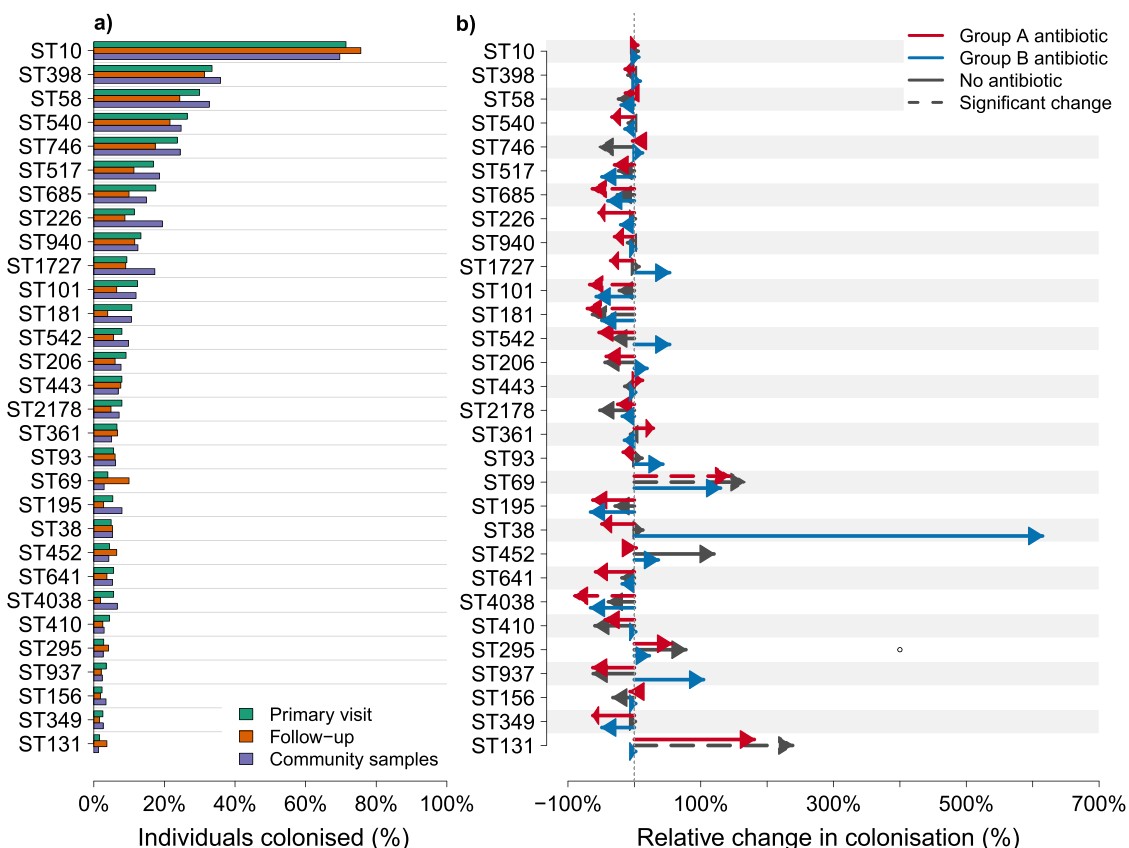

**Fig. 2 | *E. coli* colonisation rates in the community samples and in the primary and follow-up samples stratified by antibiotic type.** Panel (**a**) displays the percentage of individuals colonised (horizontal axis) by *E. coli* lineages (vertical axis) in the paired samples from the study participants recruited at the Manga Mandi hospital (labelled Primary visit or Follow-up) and for the cross-sectional community samples (labelled Community samples). Panel (**b**) displays the relative increase or decrease in colonisation in the follow-up samples stratified by type of antibiotic usage. Red lines correspond to data of participants who had received amoxicillin-clavulanate, fluoroquinolones, oral second or third generation cephalosporins, or parenteral third generation cephalosporins (nobody received other broad-spectrum parenteral β-lactams) between the primary and follow-up samples. Blue lines indicate data from participants who received another type of antibiotic. Black lines correspond to data of individuals not given antibiotics. The head of the arrow indicates a relative change in colonisation rate in the follow-up samples compared to the primary samples. Dashed arrows indicate shifts that were statistically significant at *p* < 0.10 (two-sided binomial proportions test with Benjamini-Hochberg correction for multiple testing). Both panels show values for the 31 most common lineages across all samples. Source data are provided as a Source Data file.

## The distribution of lineages circulating in Punjab differs markedly from those typically found in European and US surveillance

*E. coli* carriage in the Punjab cohorts is dominated by ST10, colonising about 70% of the individuals in the Manga Mandi hospital cohort on both sampling occasions as well as the community cohort (Fig. 2a and Supplementary Data 3). This finding is congruent with previous studies where ST10 has been found to be more frequently associated with asymptomatic colonisation than infections[6,13]. Like ST10, the second most common lineage identified, ST398, belongs to phylogroup A and is even more frequent in the community cohort. Unlike ST10, ST398 was extremely rare (3 isolates out of 3254) in a large Norwegian disease cohort[2] and in two bloodstream infection cohorts from the UK (4 out of 1448 in the BSAC1 study[1], 2 out of 718 in BSAC2[5]), as well as in asymptomatic colonisation of neonates in the UK (Mäklin et al.[6]). These data combined suggest that ST398 rarely circulates in Europe, as a stark contrast to the Punjab study population. The third most common lineage circulating in Punjab is ST58, which has also been frequently found in clinical surveillance in Western countries and was commonly isolated from meat products in a recent One Health study conducted in the US[14]. ST58 has been described as an emerging pathogenic lineage[15]. Of the other common STs found in Punjab, most are atypical in HICs. Most of our volunteers carried multiple *E. coli* lineages at the same time (median 3 in all cohorts, Fig. 3).

The overall distribution of *E. coli* STs in the two Punjabi cohorts (Fig. 2a and Supplementary Data 3) differs considerably from the distributions reported in systematic clinical surveillance studies conducted in HICs, such as Norway[2], the UK[1,5] and the US[14]. Notably, the uropathogenic ST73 and ST95, commonly observed in healthy carriage in the UK and France[4,6] are frequent in bloodstream infections in multiple HICs[2,5,14], but extremely rare in Punjab (frequency out of all BAGs 0.1% and 0.19%, respectively; Supplementary Data 3). Two other virulent sequence types ranking among the most common STs in HIC clinical surveillance, ST69 and ST131, are more frequent in Punjab than ST73 and ST95 (ST69 frequency 1.35%, ST131 frequency 0.53% in all BAGs). Notably, these are primarily found in the follow-up samples from the Manga Mandi hospital (60.56% and 57.14% of the total ST69 and ST131 identifications, respectively). Compared to the Manga Mandi hospital samples, the community samples often had higher overall *E. coli* colonisation rates with the exception of the specific clinically relevant lineages (ST69, ST131, ST648). Since these three STs were predominantly observed in the Manga Mandi hospital cohort and rarely in the community carriage, it is likely that they require additional or different selection pressure to promote their circulation among hosts.

## Use of antibiotics and hospital visit are associated with an increase in virulent lineages

Next, we compared the ST frequencies between the paired samples from the Manga Mandi cohort. In particular, we focused on the impact of antibiotics prescribed either during the first visit after the sample

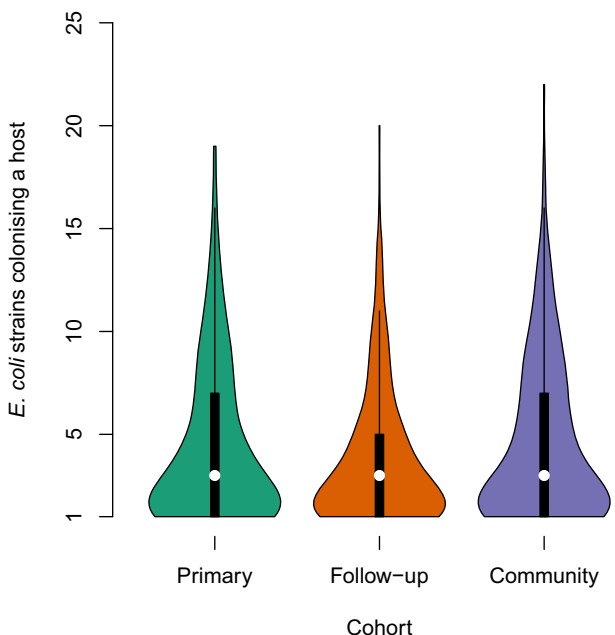

**Fig. 3 | Within-host *E. coli* diversity in the three cohorts.** This plot displays the number of *E. coli* high-quality BAGs identified within each sample as a violin plot for each of the three cohorts. Outpatients from the Manga Mandi hospital are labelled with either "Primary" (green, data presented from 427 outpatients) or "Follow-up" (orange, data presented from 431 outpatients). Community participants are labelled with "Community" (purple, data presented from 376 participants). The width of the violin corresponds to the fraction of samples containing the number of *E. coli* lineages (rows). The embedded box shows the 25–75% quantile range and the whisker shows the 75–100% quantile range. The white circle denotes the median. Source data are provided as a Source Data file.

was taken, or between the two visits, on the proportion of patients colonised by strains of each ST type (Supplementary Data 4). The antibiotics prescribed were divided into two groups. Group A comprised those considered to be most effective against *E. coli* in the gut: amoxicillin-clavulanate, fluoroquinolones, second and third generation per oral cephalosporins, and broad-spectrum parenteral β-lactams (of which only ceftriaxone was used). Group B consisted of all other systemic antibacterial medications with metronidazole, first generation cephalosporins, macrolides, and tetracyclines being the most common agents. Patients who received both a Group A and Group B antibiotic were assigned to Group A. If the patients had not been prescribed any antibiotics during the past 30-day period, their data were assigned to a 'No antibiotics' group.

The effect of antibiotics varied considerably in terms of proportions of patients colonised by the various *E. coli* lineages, but the overall trend was negative (Fig. 2b), particularly for group A antibiotics. The successful generalist lineage ST10 did not show a noticeable change for either group A or B antibiotics, while ST181 and ST685 both displayed a substantial relative reduction for group A. The most pronounced relative changes between the primary sample and follow-up sample were observed for ST69 and ST131, which were significantly enriched among those not receiving any antibiotics, and among those receiving group A antibiotics (for ST69). The estimated relative change was similarly positive for ST131 for the group A antibiotics, but the difference was not statistically significant, possibly due to a lack of statistical power under the multiple hypothesis testing correction.

### Antimicrobial treatment and hospital visit triggers competition aiding virulent *E. coli* lineages

Prompted by the changes caused by antibiotic treatment, we investigated further the competition dynamics between the *E. coli* STs in the

Manga Mandi cohort. We focused particularly on co-colonisation between the virulent lineages that were substantially enriched in the follow-up samples as described above, and other lineages which are of more commensal type, i.e. less frequently encountered in surveillance of bacteremia. An ST was defined to display substantial enrichment when more than half of its total observations were represented by the follow-up samples. Then we calculated the percentage of occurrences where these STs and the 30 most common STs were found together in the same sample, out of the total times they were found across both the primary and follow-up samples (Fig. 4).

Strikingly, the samples displaying an ST that was substantially enriched in the follow-up samples (ST69, ST131, ST405, ST504, ST648, or ST757) were significantly less diverse with respect to *E. coli* lineages present, compared to samples that contained only other, more commensal STs (one-tailed Hutcheson's *t*-test for difference in Shannon diversity indices, $p < 0.004$). Since in the study region, even the commensals generally carry a substantial number of antibiotic resistance genes, it was intriguing to find the more virulent STs capable of displacing commensals during the stress induced by antibiotic treatment. A similar, but even more pronounced exclusion effect was seen between the enriched STs, which almost never co-colonised the same individual even though they were all found to benefit from antimicrobial treatment (Supplementary Data 3 and Fig. 2b). In contrast, the more commensal STs did not exhibit strong preferences for either coexistence or competition, which is in line with the observation of the study participants typically carrying multiple lineages at the same time (Fig. 3), representing a wide range of different common STs (Fig. 2a).

### Resistance across multiple antimicrobial classes is common but carbapenem resistance remains rare

To investigate the presence of antimicrobial resistance-conferring traits in the lineages, we screened resistance genes and point mutations in each *E. coli* BAG from the 30 most common and the substantially enriched lineages (Fig. 5 and Supplementary Fig. 1) by using AMRFinderPlus. The results (Fig. 5) show the enrichment of various AMR traits in the substantially enriched lineages (STs 69, 131, 405, 504, 648, or 757) and a core set of AMR traits consisting of $bla_{TEM-1}$, $bla_{CTX-M-15}$, and gyrA_S83L and $qnrS1$ that are widespread across both the enriched and the common lineages.

We also screened the *E. coli* BAGs for alleles of genes known to confer resistance to carbapenems, which are considered the last resort antibiotics against bacterial infections caused by gram-negatives. Among the 5,247 BAGs, only 17 carried carbapenemase-encoding genes. Out of these 17, 8 were identified as the NDM-5 allele scattered across multiple STs (ST156, ST410, ST443, ST685, and ST940) (Supplementary Data 5, Fig. 5). The remaining 9 identifications were alleles of the OXA-48-like carbapenemase: a single case of OXA-48 in ST940, and 8 cases of OXA-181 in ST10, ST448 and ST58. None of the carbapenemase genes were found in two BAGs from the same sample, nor were they found in two paired samples from the same individual, suggesting that strains carrying these genes were not able to persistently colonise the study subjects in the absence of sufficient selective pressure from the use carbapenems.

### Evidence for intercontinental transmission of virulent *E. coli* lineages between Southern Asia and Northern Europe

For three STs previously identified as virulent or carrying carbapenem-resistance genes in Northern Europe (ST38, ST69, ST131), we embedded the BAGs from our study into a recombination-free phylogenetic tree (Methods) containing both carbapenemase-producing (CP) *E. coli* isolates from a population of Norwegian patients[16] and the ST-specific reference data used in this study to investigate possible transmission of these STs between Southern Asia and Northern Europe. The study found that most CP isolates were associated with travel and/or hospitalisation abroad and that 62% of the travel-associated cases were

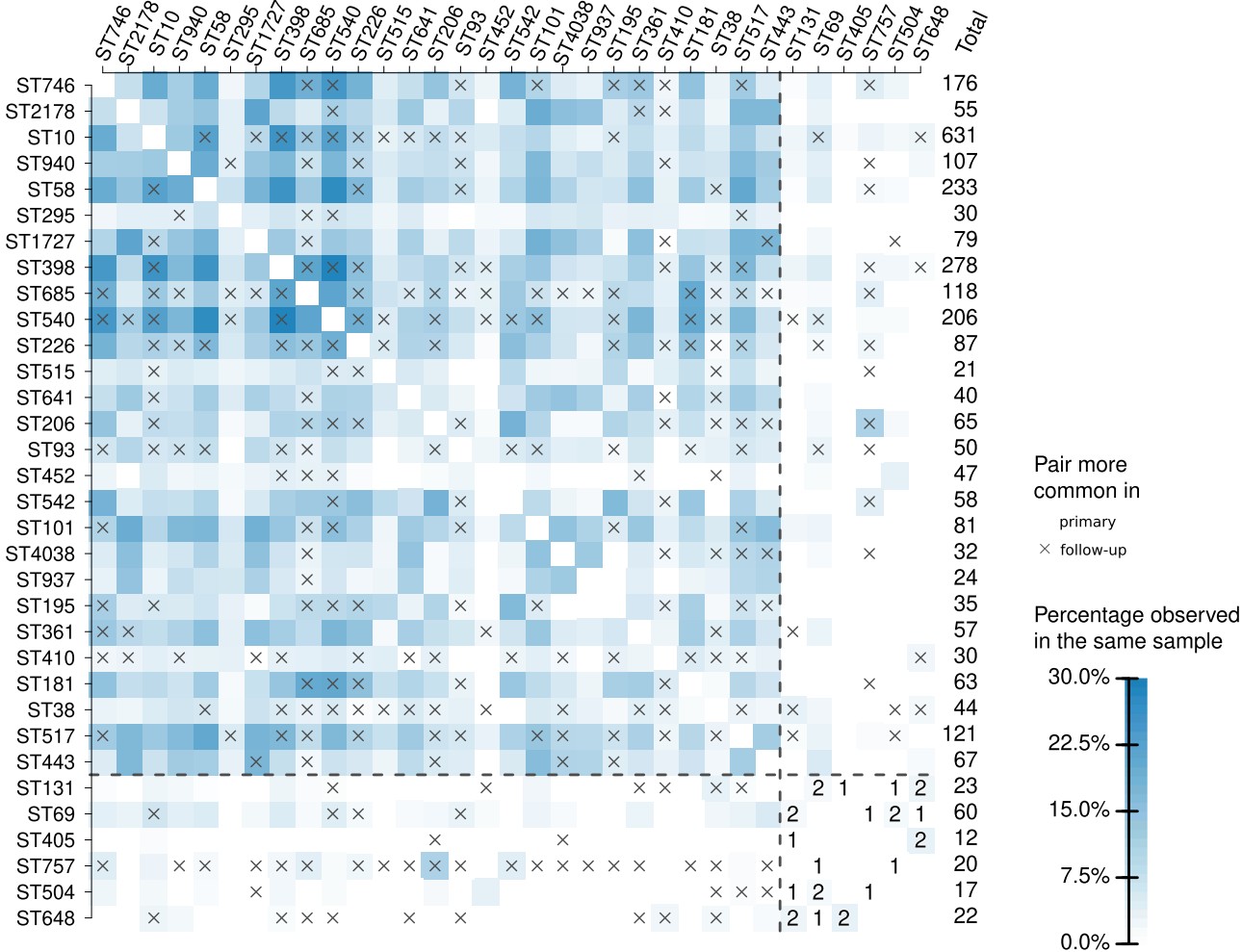

**Fig. 4 | Coexistence between *E. coli* lineages observed in the study.** The matrix shows the percentage of times an *E. coli* lineage belonging to a lineage (vertical axis) was observed together with another lineage (horizontal axis). The percentages are calculated out of the total times the lineages corresponding to the row and column were observed in the Manga Mandi hospital cohort. The column labelled *Total* shows the total number of times the lineage on each row was observed in the cohort. Lineages that had more than half of their total observations in the follow-up samples are separated by the dashed horizontal and vertical lines. For these lineages, co-observation counts are also shown in the plot. Cells labelled with an 'x' additionally denote lineage pairs more common in the follow-up samples. The plot shows the 30 most common lineages together with STs that were all substantially enriched in the follow-up samples. Source data are provided as a Source Data file.

linked to Asia, with Pakistan representing the third most common travel destination. The phylogenetic trees (Fig. 6) reveal substantial clustering of the Norwegian CP isolates with the BAGs from Punjab in all three STs, with only singleton cases of a Norwegian CP isolate aligning in a cluster containing no BAGs from our Punjab study.

Within the ST131 phylogeny (Fig. 6a), the BAGs from Punjab primarily belong to clades C0, C1, and C2, with just a few observed in clade A and none in clade B. This is congruent with previous genomic surveillance observations of ST131 clade B apparently lacking the ability to pick up MDR plasmids encoding either ESBL or carbapenemase genes, or both[2,5].

Within the ST38 phylogeny (Fig. 6b), the BAGs from Punjab form two distinct clusters, one of which shows a clear association with Norwegian CP isolates belonging to this cluster. Similarly, for ST69 (Fig. 6c), the Punjab BAGs also form two clear clusters, and only one of them contains CP isolates from Norway, however, compared with ST38, these numbers are much smaller, which limits drawing even tentative conclusions about an association.

## Discussion

Pioneering work some 40 years ago using multi-locus gel electrophoresis of enzymes painted a picture of two kinds of *E. coli* strains that could either persistently colonise the human gut and be shared by family members and unrelated individuals from different locations, or live a life as more transient passers-by[17,18]. Subsequent research has considerably refined this picture and established a DNA-based nomenclature for the increasing number of genetic lineages circulating globally, encountered in bloodstream infections, and nowadays alarmingly often carrying resistance elements to multiple classes of antibiotics[19]. Adding to this knowledge, our study is the first high-resolution genomic study of the genetic diversity of *E. coli* colonising hosts in a single location that looks beyond the MDR phenotypes and ESBL producers, and also captures the impact of antibiotic treatment on the overall diversity in healthy *E. coli* carriage.

Recent investigation into *E. coli* carriage in a cohort of neonates from the UK highlighted strong colonisation competition between strains of *E. coli* using shotgun metagenomics and reported a range of similarities and dissimilarities with the prevalence of these strains in disease surveillance in the UK and elsewhere[6]. Other recent work has examined the *E. coli* colonisation competition in an LMIC hospital setting, characterised by endemically circulating ESBL-producing Enterobacterales, with and without antibiotic treatment selection pressure[20]. These and other findings provided a further impetus for us to study colonisation and AMR diversity in a location with extensive

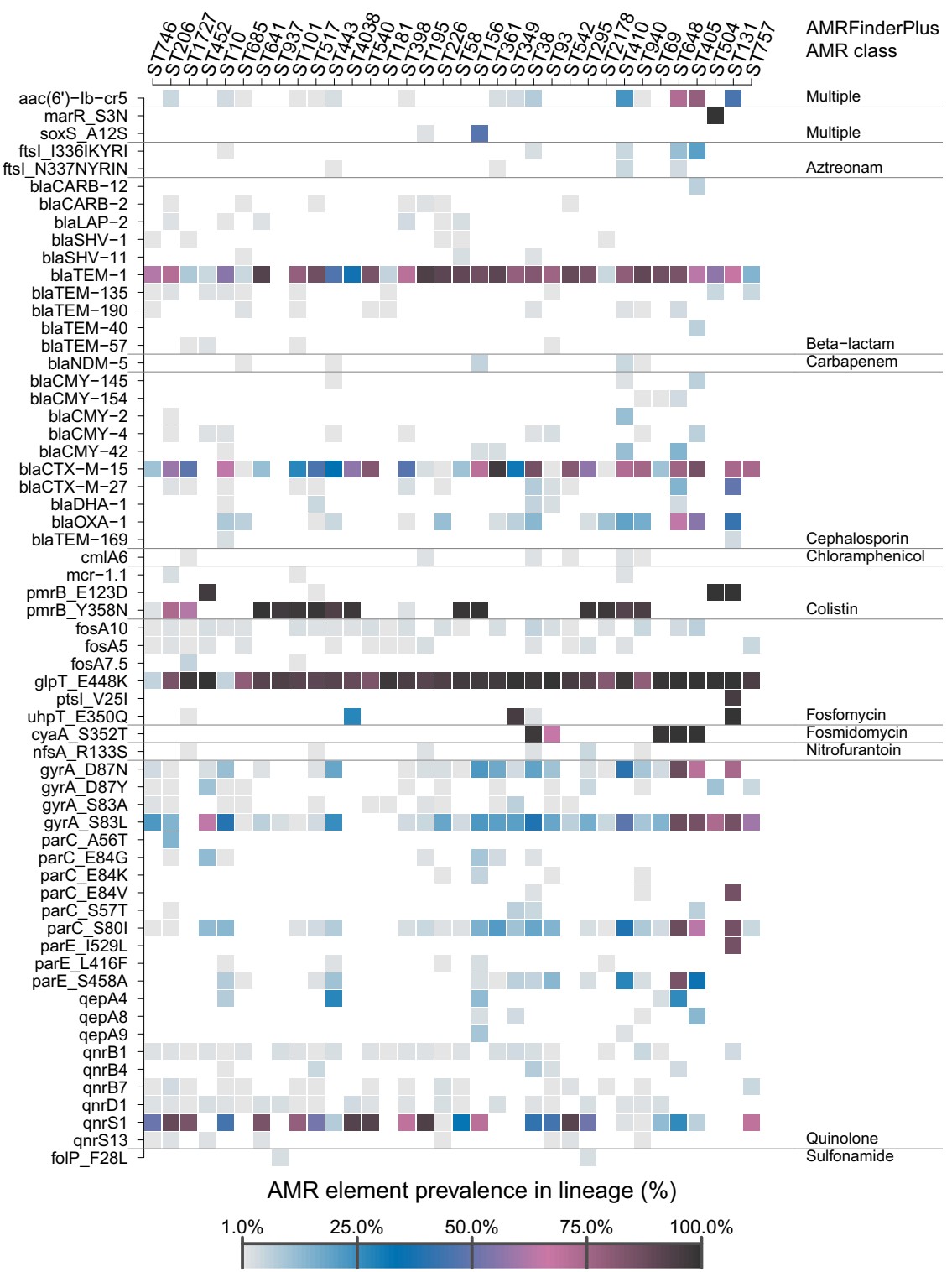

**Fig. 5 | Antimicrobial resistance determinants in the study *Escherichia coli* population.** The plot shows the percentage of times an AMR determinant (rows) was found in a specific *E. coli* lineage (columns) out of the total number of observations of the lineage. Values are included for the 30 most common lineages together with ST405, ST504, and ST543 which were all substantially enriched in the follow-up samples. The plot shows only the AMR determinants detected at least five times in the included lineages and had a minimum sequence identity of at least 95% and coverage of at least 90% to a reference in the AMRfinder+ database using the "pointx" or "allelex" method. Source data are provided as a Source Data file.

outpatient use of antibiotics, and investigate how specific genetic lineages might differentially benefit from treatments falling into distinct classes of antibiotics.

Data on medication history was limited to the Manga Mandi hospital prescription and dispensing registry since obtaining a reliable medical history from the patients themselves proved impossible. Some patients could have received antibiotics from elsewhere or bought them over the counter, but we consider it highly unlikely that any relevant quantities of antibiotics would have been acquired from other sources. Nearly all patients were very poor and all care and

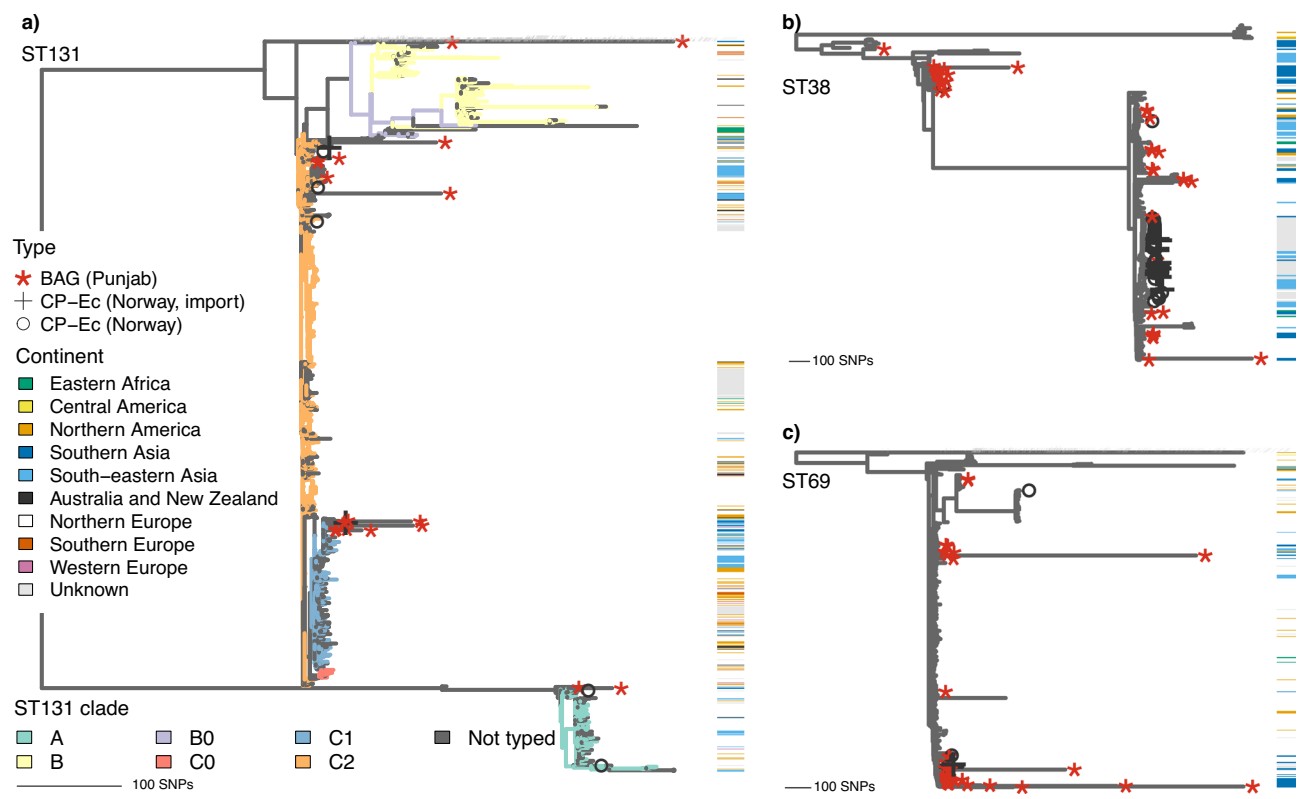

**Fig. 6 | Phylogenetic trees embedding the Punjab bin-assembled genomes into international reference sequences.** Panels contain the sequences for ST131 (**a**), ST38 (**b**), and ST69 (**c**). The tips labelled with '*' correspond to the BAGs from the current study, the tips with label '+' to carbapenemase-producing isolates from Norwegian patients with recent travel history, and the tips with the label 'o' to carbapenemase-producing isolates from Norwegian patients without recent travel history. The bars on the right-hand side of each panel denote the subcontinent (UN M49 standard) the sequence was obtained from, if available. In panel (a) the edges in the phylogeny are coloured according to the ST131 clade the leaves belong to when the clade assignments were available in the source data. Source data are provided as a Source Data file.

medications provided at the Manga Mandi hospital were part of their employment benefits, and they were allowed to visit the clinic during working hours. Thus, there was a strong counter-incentive for them to access medications from other sources, which lends support to the representativity of the medical records used in our study.

Another limitation is the storage of the stool samples of the community volunteers and the follow-up samples of the outpatients by the participants before they delivered the samples. They were instructed to collect the stool in the morning, store the sample in the shade and deliver it as early during the same day as possible. But the samples were in the jars for a few hours in ambient temperatures, which were often around 40 °C during the study period. This may have affected the bacterial composition to some degree compared to the primary samples taken by rectal swabs.

*E. coli* ST73 and ST95 represent two classical uropathogenic *E. coli* (UPEC) lineages that have been extensively studied experimentally in vitro and in vivo, and by using epidemiological surveys conducted both during the pre- and post-bacterial population genomics era[19,21–23]. A systematic comparison of the prevalence of these two lineages in neonatal colonisation with their prevalence in bacteremia surveillance indicated a heightened level of virulence but also wide circulation in the human population, as ST73 and ST95 were among the top four lineages identified in healthy babies[6]. Strikingly, our results show them a near-zero prevalence in our study region, suggesting that these two lineages exhibiting largely antibiotic-susceptible phenotypes cannot sustain themselves in a region marked by abundant antibiotic use (WHO GLASS https://www.who.int/publications/i/item/9789240062702). In contrast, ST69 and ST131, which were the other

two *E. coli* lineages in the top four of neonatal colonisation and bacteremia in the UK[5,6], often display an MDR phenotype in HICs. This is congruent with our observation of an approximately 10-fold higher prevalence than seen for ST73 and ST95. However, our data further suggest that in the absence of recent antibiotic use or visit to a clinic, ST131 and ST69 are less competitive colonisers (in Punjab, Pakistan) than other lineages, as shown by their significantly higher frequencies in the outpatient follow-up samples. This finding contradicts the recent results from a gnotobiotic mouse model related to a superior MDR ST131 colonisation ability compared with ST73 as a resident strain[24]. Since the competition may be further influenced by nutrient availability and other factors of the wider host microbiota, further research is critically needed to generate a mechanistic understanding about the interplay of intrinsic and ecological factors determining the relative successes of the top lineages in a host population.

Our observations of particular clades of the Punjab ST38 and ST69 lineages linking closely with carbapenemase-producing isolates from patients in Norway warrant further investigation. This finding corroborates the involvement of these lineages in the international transmission of carbapenemases, calling for better coordinated international efforts directed not only at disease isolates but also at asymptomatic carriage of MDR bacteria with critical resistance elements found in highly transmissible lineages. Furthermore, the tentative clustering of patient isolates primarily with particular clades seen in gut colonisation in Punjab, motivates the use of long-read sequencing technology in the future, seeking to capture plasmids and variation in their gene content in this region. Since the carbapenemase genes are typically acquired and maintained on plasmids, they may be

co-localised with virulence and other fitness-enhancing elements that increase the likelihood of observing them in disease surveillance. It is a clear clinical concern, should highly transmissible lineages of *E. coli* follow the same path of convergence between AMR and virulence that has been observed in *Klebsiella pneumoniae*[25,26].

ST58 has been found relatively frequently in human bacteremia in both Norway[2] and the UK[5] A recent systematic One Health survey from the U.S. clearly attributed these infections to foodborne illness originating from meat[14], in line with a previous reporting of ST58 as a primarily animal-adapted lineage with high zoonotic pathogenic potential[27]. The very high prevalence of ST58 in human gut colonisation in Punjab (the third most common lineage) indicates that either the zoonotic transmission pressure is very high in this region, or the lineage is well adapted to persistently colonising and transmitting also among humans, in contrast with the evidence from HIC clinical and meat surveillance data. It is further possible that the baseline microbiota composition of the hosts differs systematically to benefit ST58 in colonisation competition. Further research is thus warranted into the molecular mechanisms allowing its persistence in the human gut, to complement recent findings on plasmid content stratifying the distribution of ST58 across different animal hosts[15] and on the lineage-specific associations between virulence genes within the ST58 population[28].

The finding of ST398 being the second most frequent coloniser of the study population is intriguing since it has been very rarely identified in either disease, in healthy colonisation or in food sources in Western countries, based on the large genomic surveillance studies mentioned above (Gladstone et al. [2]; Pöntinen et al. [5]; Mäklin et al. [6]; Liu et al. [14]). Lack of genomic resolution disease surveillance data from the study region currently prevents drawing conclusions about the relative virulence potential of ST398, however, there is a clear need for deeper investigation into the ecological and genetic factors that underpin its remarkably higher success in colonisation and transmission in this region.

We found a wide diversity of genetic lineages harbouring extensive levels of AMR, which is hardly surprising, given the considerable frequency of travellers from Europe to Asia becoming colonised with MDR *E. coli*[7,29,30]. The most common lineage in our study, ST10, colonised approximately the same fraction of patients visiting the clinic as participants in the community arm of the study. In addition to several other classes of resistance elements, ST10 was detected to carry multiple different versions of carbapenemase-producing genes (both NDM and OXA). While the prevalence of these genes was not particularly high, usage of carbapenems is relatively rare in this region due to their cost (WHO GLASS: https://www.who.int/publications/i/item/9789240062702), which suggests that generalist lineages such as ST10 are capable of acquiring and maintaining these genes without an excessive fitness cost. Given its ability to effectively colonise the human gut asymptomatically, ST10 has the potential to act as a global dissemination vehicle of AMR genes without becoming notified as a prevalent source, since beyond some exceptions considering community carriage[31], surveillance is nearly exclusively performed from clinical samples, in particular for critical AMR such as ESBLs and carbapenemases[16,32]. Neither the global population structure of ST10, nor the molecular mechanisms behind its general success in colonising humans are currently well understood, which calls for in-depth studies of this generalist *E. coli* to aid future efforts in mitigating the public health burden across the wide spectrum of extra-intestinal pathogenic *E. coli* genetic variation.

## Methods

### Ethics approval
The study was approved by the ethics committee of the School of Biological Sciences, University of the Punjab (ref no: SBS/822/15). Participants were given information about the study in Urdu, and written informed consent was obtained from both the outpatients and the community volunteers.

### Sample collection
Outpatients were recruited at the Punjab Social Security Health Management Company Hospital Raiwind (Manga Mandi hospital) in the Lahore Metropolitan area between April 2016 and June 2016. It provides primary and secondary healthcare for 83,000 industrial workers and their families. Patients of all ages visiting the clinic were referred to us by their physicians. After obtaining informed consent, the patients were interviewed. Their rectal swabs were taken with nylon swabs (FLOQSwab™, Copan Diagnostics, Italy) and stored in a modified Cary-Blair medium (FecalSwab™, Copan Diagnostics, Italy). The specimens were cultured within 48 h on CLED agar at 37 °C, and all growth was collected after 48–72 h and frozen at -80 °C in milk glycerol. The patients were subsequently handed out jars for follow-up stool samples to be delivered after four weeks. The returned jars were stored in cold boxes, the stools were transferred to FecalSwab™ tubes the same evening, and the samples were then processed the same way as the previous ones.

In addition to the Manga Mandi Hospital outpatients, community volunteers were recruited at eight different locations in the Punjab region, covering both urban and rural settings between March 2016 and June 2016. After interviewing and obtaining informed consent from the volunteers, they were given jars for collecting a stool sample to be delivered to us by the following morning. These stool samples were stored and processed identically to the outpatients' follow-up samples. Neither the outpatients nor the community received compensation for taking part in the study. The sample collection locations are presented in Fig. 1a and the overall sample processing is visualised in Fig. 1b.

### Sequencing and population genomics analyses
Genomic DNA was extracted from bacterial growth on a Cystine Lactose Electrolyte-Deficient (CLED) agar plate (Thermo Fisher Scientific, Waltham, MA, USA) using the ZymoBIOMICS 96 MagBead Kit (Zymo Researcher, CA, USA) and sequenced en masse at high depth using Illumina NovaSeq 6000 machines. Sequencing generated an average of 4,420,020 150 bp paired-end reads, equating to a coverage depth of 93X for samples dominated by E. coli.

### Species-level taxonomic profiling and binning
We used the mGEMS pipeline[12] as described in a previous study[6] to identify the *E. coli* lineages in each sequencing sample. Briefly, the sequencing reads were first preprocessed with fastp[33] (v0.23.2, OpenGene/fastp, we ran quality filtering, adaptor and polyG tail trimming, and paired-end read error correction) and pseudoaligned against the species-level index of the 661k genomes collection[34] produced as part of another study[35] (https://doi.org/10.5281/zenodo.7736981) with Themisto[35] (v3.0.0-rc). Next, the pseudoalignments were used to estimate species-level relative abundances using mSWEEP[11] (v2.0.0, PROBIC/mSWEEP) and the reads were assigned to species-level bins with the mGEMS binning algorithm[12] (ran from mSWEEP v2.0.0 using the --bin-reads option, described in). Bins were created for species with a relative abundance of at least 0.1%.

### *E. coli* index construction and lineage identification
Before performing lineage identification, we expanded the *E. coli* index from[6] (available from https://doi.org/10.5281/zenodo.6656881) with isolate assemblies from four large studies conducted in Laos, Pakistan, Thailand, and Malawi[10,36–38]. Because these studies added a significant number of assemblies from lineages that were underrepresented in the original index, we redefined the lineages in the expanded index using PopPUNK[39] (v2.5.0, we used the same parameters as in a previous study[6]). The expanded index files are available from Zenodo (https://doi.org/10.5281/zenodo.10077625).

### Lineage-level taxonomic profiling and binning
After constructing the expanded index, we pseudoaligned reads from each *E. coli* bin against the index with Themisto (v3.0.0-rc) and

estimated lineage-level relative abundances using mSWEEP (v2.0.0). The reads were then assigned to lineage-level bins using mGEMS for lineages with a minimum relative abundance of at least 1%, and a set of in-house scripts (tmaklin/coreutils_demix_check, v0.3.2) was used to check that the reads assigned to each lineage-level bin had good representation in the expanded index. With these scripts, a bin was considered to pass the check if the mash[40] (v2.3) distance between the reads in the lineage-level bin and the assemblies in the index was lower than the maximum distance between the assemblies in the index (score 1 from the in-house scripts) or below an acceptable threshold slightly above the maximum (score 2 from the in-house scripts).

### Assembly from lineage-level bins and quality filtering
Reads from each lineage-level bin created in the previous step were assembled using shovill (v1.1.0, tseemann/shovill, read correction disabled, minimum contig length 300, minimum coverage 1.0x) which is an assembly pipeline built around the SPAdes[41] assembler (we used v3.15.5). Each bin-assembled genome (BAG) was subjected to further quality control by calculating the completeness and contamination using checkm[42] (v1.2.1, taxonomy workflow for *E. coli*). BAGs with a completeness score below 90% or contamination score above 10% were removed. Assembly statistics were calculated for the remaining BAGs using assembly stats (commit id c006b9c, sanger-pathogens/assembly-stats) and assemblies shorter than 4 Mb or longer than 6 Mb in total length were removed. A total of 5,671 BAGs (out of an initial total of 6,370 lineage-level bins) passed these filters and were considered high-quality identifications.

### Antimicrobial resistance gene and allele detection
We ran AMRFinderPlus[43] (v3.10.42, database version 2022-10-11.2, *Escherichia* taxonomy group, minimum identity 95% and minimum coverage 90%) on the high-quality BAGs and extracted the results that were assigned to the "AMR" class with the ALLELEX or the POINTX methods.

### Quality filtering for phylogenies
Extra-quality filtering was performed on the 5,671 BAGs in order to determine the assemblies that were of suitably high quality to include in a phylogenetic analysis. We used the contig-level analysis in gunc[44] (v1.0.5, with modifications added in tmaklin/gunc, database version 2.1, --sensitive flag enabled) to identify contigs that did not have any genes assigned to the *Escherichia* taxonomy and removed these contigs using an in-house script (available from tmaklin/bioinfo-scripts). Afterwards, assemblies shorter than 4 Mb or longer than 6 Mb, and assemblies with more than 600 contigs (after filtering with gunc) were removed. We ran gunc again on the remaining BAGs to identify chimerism and retained those that passed the default filter. In total, these extra filters left 3,116 BAGs for the phylogenetic analyses.

### Combined isolate and BAG phylogenies
Within each *E. coli* lineage included in the phylogeny analyses, we first determined a reference assembly for each lineage by running assembly stats on the isolate assemblies and designating the assembly with the highest median contig lengths (N50) within each lineage as the reference. We then mapped all other isolate assemblies from each lineage against their respective references using SKA2[45,46] (v0.3.3, bacpop/ska.rust) and ran gubbins[47] (v3.2.1, nickjcroucher/gubbins, --best-model option enabled and maximum 15% missing sites allowed) on the resulting alignment to determine recombination sites. A custom script (available from tmaklin/bioinfo-scripts) was used to extract the recombination sites identified by gubbins from the alignment.

Next, SKA2 (*k*-mer size 31) was used to index all isolate assemblies and BAGs belonging to the same *E. coli* lineage that passed the extra-quality filtering described earlier. We then used SKA2 to remove the sites affected by recombination to build reference-free alignments

containing both the isolate assemblies and BAGs. Finally, a tree was constructed from the reference-free alignment by running snp-sites[48] (commit id 52c98cb, sanger-pathogens/snp-sites) on the alignment and using VeryFastTree[49] (v3.3.0, citiususc/veryfasttree, GTR+Gamma model with max. 20 categories) to infer a maximum likelihood phylogeny.

### Reporting summary
Further information on research design is available in the Nature Portfolio Reporting Summary linked to this article.

## Data availability
The sequencing data used in the study is available from the ENA as part of project accession number PRJEB36642 and a list of the run accession numbers that originated from this study is included in Supplementary Data 1. The BAGs created in this study are available from Zenodo (https://doi.org/10.5281/zenodo.10075671) and their associated meta-data and assembly quality statistics are available in Supplementary Data 2 and Supplementary Data 3. The indexes used in the species and lineage-level analyses are available from Zenodo (species index from https://doi.org/10.5281/zenodo.6656881, *E. coli* lineage index from https://doi.org/10.5281/zenodo.10077625). Source data are provided with this paper.

## Code availability
A git repository containing the scripts and source data files used for creating the figures in the manuscript is available at themaklin/pakistan-e_coli-diversity-plots. Exact programme calls and options used to run the analyses are available as Slurm batch job scripts from Zenodo (https://doi.org/10.5281/zenodo.10077811). All software used in the study are freely available from sources indicated in the Methods section.

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

## Acknowledgements

The authors wish to acknowledge CSC – IT Center for Science, Finland, for computational resources. We wish to thank the staff of the Punjab Social Security Health Management Company Hospital Raiwind for allowing us to recruit the outpatients. We are deeply grateful to all the volunteers for their willingness to participate. This study was funded by VTR grants of Helsinki University Hospital [TYH 2012141, TYH 2013218 and TYH 2014216, AK], the Sigrid Jusélius Foundation [1726, AK], Trond Mohn Foundation (BATTALION grant, JC, RAG, AKP, ØS), Wellcome Trust Grant 206194, and Academy of Finland (EuroHPC grant, JC, TM, AMRIWA grant, AK).

## Author contributions

Study design TKh, JC, AK; data collection TKh, AK; Laboratory work TKh, SM, MI, MWA; Statistics and bioinformatics TM,TKa, JC; Visualisations TM; Drafting of the manuscript TKh,TM,TKa,JC,AK; critical comments on manuscript RAG,AKP, SM, HAT, ØS, JP, MI, MWA; approving final manuscript all. JC and AK contributed equally to the publication.

## Competing interests

The authors declare no competing interests.
