## [Peer Review File · Nature Communications]

Deep sequencing of *Escherichia coli* exposes colonisation diversity and impact of antibiotics in Punjab, PakistanREVIEWER COMMENTS

Reviewer #1 (Remarks to the Author):

The paper of Khawaja et al. presents very interesting data on the carriage of E. coli strains in populations of Punjab, Pakistan. These data are original because (i) they are rare outside HIC and (ii) the deep sequencing approach allows to capture the intra-individual diversity. As expected from the previously published data, the population structure is different from the HIC. Overall, the work is well done and the data are clearly presented. I have in fact very few comments on the manuscript.

I am surprised that in such a huge cohort of individuals sampled in Pakistan, only two authors among 13 originate from Pakistan, knowing the amount of work that it represents.

Lines 322_323, the first description of the primarily animal character and human emergence of the ST58 is from <https://doi.org/10.1093/molbev/msv280> as it belongs to the CC87 using the Pasteur Institute MLST scheme.

Reviewer #2 (Remarks to the Author):

Khawaja et al utilised deep sequencing of plate sweeps to describe the diversity in E. coli colonisation and the impact of antibiotic use in Punjab, Pakistan. The authors generated 5,247 high-quality E. coli genomes (assembled from binned reads) using samples derived from (i) 494 hospital patients (with paired control and primary samples) and (ii) 423 volunteers representing the community samples. The short-read sequencing effort employed in this study generated data resulting in an average of 93x depth of coverage per sample that was dominated by E coli. The key findings uncover the diversity of sequence types within hosts, which has important implications in understanding ecological interactions, and pathogen and AMR transmission dynamics.

The study produced high quality data and the analysis is comprehensive and well executed. The motivation for the study is also clearly stated. I commend the authors for their efforts here. I have no major concerns as I believe the computational approaches are appropriate, and the sample types and size are adequate to answer their key objectives, which was to quantify the within-host genetic diversity at strain level, assess the impact of antibiotic use on this diversity, and draw comparisons to genomic surveillance data collected from Norway, UK, and the US. Additionally, the authors clearly stated their findings and are transparent in acknowledging their studies limitations where appropriate - e.g., lines 244 - 245: "...which limits drawing even tentative conclusions about an association."

Some minor comments for consideration:

Line 59: I would suggest replacing "[entire] genetic diversity of E. coli in their samples" with something to the effect of "[a more complete] genetic diversity of E. coli in their samples". Plate sweeps are limited to culturing which has a bias towards dominant strains. Whole-community shotgun metagenomic sequencing of clinical samples at depth has in the past been shown to detect sequence types previously not detected by culturing.

Figure 1A - White text of cities on the map is difficult to read.

Figure 1B - The workflow on the right hand side community samples is not entirely clear, as the section starting "outpatients asked to self collect..." merges to the righthand side. Perhaps some stylistic edits to clarify (more than just arrows) e.g., boxed outline and justified text(?).

Figure 3 - purple violin for community samples seems to extend beyond the max value of the y-axis

Figure 5 - some of the lighter coloured squares are difficult to discern. may be worthwhile having borders/outline

Figure 6 - some white space between panels A, B, and C would help with clarity of presentation.

Perhaps outside the scope of the current study to include any analyses, but can the authors discuss briefly the potentials of:

- The role of the baseline microbiota composition that may be predisposing individuals in Punjab compared to Norway, UK, and US with respect to the prevalence of ST58?
- The role of prophages in conferring community vs., hospital ST prevalence. e.g., do hospital STs carry more prophages?
- Endemicity of clinically relevant clones from Europe not seen in Pakistan - other virulence factors? A recent study of blood isolates of E. coli demonstrated an increased association of bacterial cells with serum cholesterol. Can the authors touch on what genomic factors of ST398 might make it commensal in Pakistan but clinically relevant in Norway/UK – are there geographically differences between the genomes of ST398 in Pakistan compared to Norway/UK?

Some very minor editorial edits throughout the manuscript, for example:

- uniform font and formatting of text.
- line 74: "... presents the first opportunity to not only [to] quantify the within-host genetic diversity..."
- "... presents the first opportunity to not only quantify the within-host genetic diversity..."

Reviewer #3 (Remarks to the Author):

What are the noteworthy results?

- The scale of the study (number of samples from the outpatients and community) and number of isolates have not been collected earlier in Pakistan. The within host diversity of E. coli and different strains is an important finding with respect to surveillance perspective.

Will the work be of significance to the field and related fields? How does it compare to the established literature? If the work is not original, please provide relevant references.

- The work is important from a surveillance point of view that gives a broad background of E. coli strains present in Pakistan outpatients and community members from the Punjab province. There have been very few genomic studies from Pakistan, so this study adds to this literature.

Does the work support the conclusions and claims, or is additional evidence needed?

- I am not convinced with few of the claims and conclusions made by the author. The methods is not completely sound with many gaps as I have highlighted later line by line.

Are there any flaws in the data analysis, interpretation and conclusions? Do these prohibit publication or require revision?

- The figure 1 needs some clarification. Also the interpretation should be more guarded given the limitations in collecting the stool samples in a cold, controlled environment.

Is the methodology sound? Does the work meet the expected standards in your field?

- The methodology has few missing points especially on the survey form that was collected, how it was collected (language used), etc. Was a survey collected from the community participants, were they reimbursed for their time/effort? The timeline of data collection (only stated in supplementary data) is not mentioned in the main draft. The bioinformatics portion seems very vast and well done.

Is there enough detail provided in the methods for the work to be reproduced?

Yes, I guess there is enough details provided.

Also below are the points (line by line) that could improve the draft.

1. Line 48: resistant pandemic clones 'pose' a severe clinical concern.
2. Line 53: why are certain words in another font? Same with line 65, 154-156
3. Line 57: Laos study on diversity within one ecoli
4. Line 74: to not only 'to' quantify. Delete one 'to'

5. Line 84: what is 'there'? specify 'Pakistan'
6. Line 88: define 'control' sample. Why did you collect this sample? Does it imply that carriage of E. coli strains in gut disturbed by antibiotics administration will return to normal within a month? You do not provide any reasoning for this?
7. Line 96: unclosed bracket after mSWEEP/mGEMS tools
8. Figure 1: the 994 outpatients included from Manga Mandi Hospital, were the patient's data discarded if they did not follow up with stool sample after 1 month.
 - How was the storage of the stool sample before the collection and storage in cold boxes, by the volunteer participants and outpatients contacted after a month.
 - The total 1411 comes from 2 samples from 494 outpatients ($2 \times 494 = 988$) and 1 sample from 423 participants. However, the figure 1b is misleading as it states that 2 rectal swabs were taken? Why was only one rectal swab stored at 4 degree Celsius and cultured? Please clarify this point.
9. Line 115 : ST10 has been identified living a more 115 commensal lifestyle: kindly rephrase to make the sentence more objective and scientific. (for e.g. ST10 has been identified more commonly in samples from non-diseased participants than clinical samples.)
10. Line 118: use a better adverb clause instead of 'in contrast to ST10', explain it in a better sentence or sentences.
11. Line 121: If the rarity of ST398 corroborates the commensal nature of this lineage, you need to confirm this from a community cohort in Europe and cannot imply that it rarely circulates in Europe. Also you stated later in line 178/179 - 'lineages which are of more commensal type, i.e. less frequently encountered in surveillance of 179 bacteremia', how can you imply this based on only Europe data? You need to have clinical surveillance data from other parts of the world, including South Asia to apply this reason.
12. Line 123: 'After ST10 and ST398, the third most common lineage circulating in Punjab is ST58, which is also frequently found in clinical surveillance in Western countries and was commonly isolated from meat products in a recent One Health study conducted in the U.S.' Why did you use 'also' when ST398 is rarely found in clinical studies in the Europe?
13. Line 131: inconsistent usage of the term - HIC sometime mentioned high-income countries
14. Line 134: above cited studies - kindly cite again as it is ambiguous which above study you are talking about
15. Line 136: a very long and difficult to read sentence. Kindly rephrase
16. Line 144: check indicates spelling
17. Line 143: Kindly explain the sentence in a better way. Why is difference in ST69, ST131 and ST648 carriage rates in Hospital and community samples indicative of lower frequencies of sustained carriage? (in which population, why?)
18. Line 149 : when were the antibiotics administered? Before/during or after the participation?
19. Line 158 - use 'being' instead of 'as' the most common agents
20. Line 164: what do you mean by generalist?
21. Line 170: use either 'similarly' or 'also'
22. Line 180: seems like an incomplete sentence: 'We defined 'missing' as a lineage...'
23. Line 256-258 - 'captures the impact of antibiotic treatment on the overall diversity in healthy carriage'. This statement is not clear as the criteria of the outpatients being referred for the study is not stated. Also, timing of the antibiotic administration is not stated. Especially when outpatients can easily access antibiotics without prescription. Hence we cannot surmise whether this diversity is in 'healthy' carriage.
24. Line 319 - the sentence is too long and difficult to understand
25. Line 391 - (v3.0.0-rc,) remove the comma
26. Line 385-396 - Methods - inconsistency in referencing with sometimes the reference appearing within a bracket, before a bracket, after a comma in a bracket)
27. No mention of interview questionnaire (survey) for the participants
28. No mention of the timeline and year of recruitment and processing of samples (except in the metadata in the supplementary table)
29. No mention of limitation of collection and storage of stool samples from community and their implications on data quality

Response to reviewer comments

Reviewer #1 (Remarks to the Author):

The paper of Khawaja et al. presents very interesting data on the carriage of *E. coli* strains in populations of Punjab, Pakistan. These data are original because (i) they are rare outside HIC and (ii) the deep sequencing approach allows to capture the intra-individual diversity. As expected from the previously published data, the population structure is different from the HIC. Overall, the work is well done and the data are clearly presented. I have in fact very few comments on the manuscript.

We thank the reviewer for their time and the kind words.

I am surprised that in such a huge cohort of individuals sampled in Pakistan, only two authors among 13 originate from Pakistan, knowing the amount of work that it represents.

Please note that the first author is a Finnish-Pakistani medical doctor who is a certified specialist in infectious diseases, who designed and supervised the study on-site in Punjab. The samples were collected and participants interviewed by TK with the help of two local nurses and a biochemist. The contribution of the biochemist and the nurses is not enough to fulfil the journal's policy for authorship. The two Pakistani collaborators contributed to the handling and culture of the samples and not the sampling per se.

Lines 322_323, the first description of the primarily animal character and human emergence of the ST58 is from <https://doi.org/10.1093/molbev/msv280> as it belongs to the CC87 using the Pasteur Institute MLST scheme.

Thank you for pointing out the relevant literature, we have added a citation to the suggested study.

Reviewer #2 (Remarks to the Author):

Khawaja et al utilised deep sequencing of plate sweeps to describe the diversity in *E. coli* colonisation and the impact of antibiotic use in Punjab, Pakistan. The authors generated 5,247 high-quality *E. coli* genomes (assembled from binned reads) using samples derived from (i) 494 hospital patients (with paired control and primary samples) and (ii) 423 volunteers representing the community samples. The short-read sequencing effort employed in this study generated data resulting in an average of 93x depth of coverage per sample that was dominated by *E. coli*. The key findings uncover the diversity of sequence types within hosts, which has important implications in understanding ecological interactions, and pathogen and AMR transmission dynamics.

The study produced high quality data and the analysis is comprehensive and well executed. The motivation for the study is also clearly stated. I commend the authors for their efforts here. I have no major concerns as I believe the computational approaches are appropriate, and the sample types and size are adequate to answer their key objectives, which was to quantify the within-host genetic diversity at strain

level, assess the impact of antibiotic use on this diversity, and draw comparisons to genomic surveillance data collected from Norway, UK, and the US. Additionally, the authors clearly stated their findings and are transparent in acknowledging their studies limitations where appropriate - e.g., lines 244 - 245: "...which limits drawing even tentative conclusions about an association."

Thank you for the review, we are happy to hear that our study was perceived well. We have responded in detail to the minor comments below.

Some minor comments for consideration:

Line 59: I would suggest replacing "[entire] genetic diversity of *E. coli* in their samples" with something to the effect of "[a more complete] genetic diversity of *E. coli* in their samples".

We have changed this sentence to "[capture a more complete picture of the] genetic diversity of *E. coli* in their samples"

Plate sweeps are limited to culturing which has a bias towards dominant strains. Whole-community shotgun metagenomic sequencing of clinical samples at depth has in the past been shown to detect sequence types previously not detected by culturing.

We agree that in principle whole-community shotgun metagenomic sequencing would be (even) less biased than plate sweeps but in practice its applicability to analysing the within-host diversity of a particular species is limited because much of the sequencing capability will be spent on off-target species. In fact, in some studies metagenomic sequencing has failed to identify ESBL-producing *E. coli*s found by culture from the same stool sample. Particularly in our study, where the median number of *E. coli* lineages detected in a single host at a time point was 3, untargeted sequencing without prior enrichment by culture would likely have limited capability to reliably detect minority variants colonising a host, as the relative abundance of *E. coli* in the healthy human gut is typically low.

Figure 1A - White text of cities on the map is difficult to read.

We've changed the colour scheme and increased the font size slightly to improve the readability of the place names on the map.

Figure 1B - The workflow on the right hand side community samples is not entirely clear, as the section starting "outpatients asked to self collect..." merges to the righthand side. Perhaps some stylistic edits to clarify (more than just arrows) e.g., boxed outline and justified text(?).

We've changed the layout and clarified the contents to make the workflow clearer.

Figure 3 - purple violin for community samples seems to extend beyond the max value of the y-axis

We've increased the range of values on the y-axis.

Figure 5 - some of the lighter coloured squares are difficult to discern. may be worthwhile having borders/outline

We've changed the scheme so that the lighter squares are replaced with a more discernible colour for non-zero values.

Figure 6 - some white space between panels A, B, and C would help with clarity of presentation.

Thank you for the suggestion, we've increased the amount of white space between the panels.

Perhaps outside the scope of the current study to include any analyses, but can the authors discuss briefly the potentials of:

- The role of the baseline microbiota composition that may be predisposing individuals in Punjab compared to Norway, UK, and US with respect to the prevalence of ST58?:

This is a relevant hypothesis and we have added a brief comment about it in the Discussion.

- The role of prophages in conferring community vs., hospital ST prevalence. e.g., do hospital STs carry more prophages?

This is an interesting question but given the enormous diversity and rapid evolution of prophages, it would be necessary to have long-read sequence data to do a robust analysis of such an association.

- Endemicity of clinically relevant clones from Europe not seen in Pakistan - other virulence factors? A recent study of blood isolates of E. coli demonstrated an increased association of bacterial cells with serum cholesterol. Can the authors touch on what genomic factors of ST398 might make it commensal in Pakistan but clinically relevant in Norway/UK – are there geographically differences between the genomes of ST398 in Pakistan compared to Norway/UK?

We are sorry if this was unclear in the manuscript, but ST398 is extremely rare in the European BSI cohorts and also in a large healthy colonisation cohort study done on UK neonates and in a One Health study in the U.S. Since only a handful of ST398 isolates have been observed among the >5,000 genomes from the two European cohorts there is unfortunately no statistical power to do comparative genomics analysis between these and the genomes obtained from Pakistan. However, as a reflection on the comments from other reviewer, we have edited this section to avoid misinterpretation and added a brief comment on ST398 findings in the Discussion.

Some very minor editorial edits throughout the manuscript, for example:

- uniform font and formatting of text.

It is unclear what caused the appearance of different fonts in places. However, such variability will be removed in the typesetting process of the journal.

- line 74: "... presents the first opportunity to not only [to] quantify the within-host genetic diversity..."

"... presents the first opportunity to not only quantify the within-host genetic diversity..."

Fixed.

Reviewer #3 (Remarks to the Author):

What are the noteworthy results?

- The scale of the study (number of samples from the outpatients and community) and number of isolates have not been collected earlier in Pakistan. The within host diversity of E. coli and different strains is an important finding with respect to surveillance perspective.

Will the work be of significance to the field and related fields? How does it compare to the established literature? If the work is not original, please provide relevant references.

- The work is important from a surveillance point of view that gives a broad background of E. coli strains present in Pakistan outpatients and community members from the Punjab province. There have been very few genomic studies from Pakistan, so this study adds to this literature.

Does the work support the conclusions and claims, or is additional evidence needed?

- I am not convinced with few of the claims and conclusions made by the author. The methods is not completely sound with many gaps as I have highlighted later line by line.

Are there any flaws in the data analysis, interpretation and conclusions? Do these prohibit publication or require revision?

- The figure 1 needs some clarification. Also the interpretation should be more guarded given the limitations in collecting the stool samples in a cold, controlled environment.

Is the methodology sound? Does the work meet the expected standards in your field?

- The methodology has few missing points especially on the survey form that was collected, how it was collected (language used), etc. Was a survey collected from the community participants, were they reimbursed for their time/effort? The timeline of data collection (only stated in supplementary data) is not mentioned in the main draft. The bioinformatics portion seems very vast and well done.

Is there enough detail provided in the methods for the work to be reproduced?

Yes, I guess there is enough details provided.

We thank the reviewer for their comments and have responded in detail below.

Also below are the points (line by line) that could improve the draft.

1. Line 48: resistant pandemic clones 'pose' a severe clinical concern.

Rephrased: *"the frequent emergence of novel virulent and resistant pandemic **represent** a severe clinical concern"*.

2. Line 53: why are certain words in another font? Same with line 65, 154-156

We are uncertain about the root cause of this, it may have to do with file format conversion, however, any such irregularities will be removed by the journal typesetting process.

3. Line 57: Laos study on diversity within one ecoli

The Laos study that we cite in our manuscript was conducted using isolate data so it is not comparable to our approach.

4. Line 74: to not only 'to' quantify. Delete one 'to'

Fixed.

5. Line 84: what is 'there'? specify 'Pakistan'

Changed to: *"important clones in Europe sporadically turn up in Pakistan but do not become endemic in this region"*.

6. Line 88: define 'control' sample. Why did you collect this sample? Does it imply that carriage of E. coli strains in gut disturbed by antibiotics administration will return to normal within a month? You do not provide any reasoning for this?

We agree that the use of 'control sample' here may be misleading and have changed the term to 'follow-up' sample throughout the manuscript, which is better reflective of the nature of the sample.

7. Line 96: unclosed bracket after mSWEEP/mGEMS tools

Fixed.

8. Figure 1: the 994 outpatients included from Manga Mandi Hospital, were the patient's data discarded if they did not follow up with stool sample after 1 month.

Yes, the data were not included in the analyses if the follow-up sample had not been returned. We have clarified this in Figure 1b.

- How was the storage of the stool sample before the collection and storage in cold boxes, by the volunteer participants and outpatients contacted after a month.

Participants were asked to collect the sample in the morning and return them preferably before noon, but latest by 3PM. They were asked to store them in shade. We added this information to the manuscript (please see the final comment)

- The total 1411 comes from 2 samples from 494 outpatients ($2 \times 494 = 988$) and 1 sample from 423 participants. However, the figure 1b is misleading as it states that 2 rectal swabs were taken? Why was only one rectal swab stored at 4 degree Celsius and cultured? Please clarify this point.

We collected two rectal swabs. The first one was for cultures presented here and the second was collected in eNat tubes to enable later PCR analyses, which were not used in this study. We've removed mention of the second swab from the figure to make this clear.

9. Line 115 : ST10 has been identified living a more 115 commensal lifestyle: kindly rephrase to make the sentence more objective and scientific. (for e.g. ST10 has been identified more commonly in samples from non-diseased participants than clinical samples.)

Rephrased as suggested.

10. Line 118: use a better adverb clause instead of 'in contrast to ST10', explain it in a better sentence or sentences.

This has been rewritten.

11. Line 121: If the rarity of ST398 corroborates the commensal nature of this lineage, you need to confirm this from a community cohort in Europe and cannot imply that it rarely circulates in Europe. Also you stated later in line 178/179 – 'lineages which are of more commensal type, i.e. less frequently encountered in surveillance of 179 bacteremia', how can you imply this based on only Europe data? You need to have clinical surveillance data from other parts of the world, including South Asia to apply this reason.

This is a justified criticism and we have carefully rephrased the section to remove the sloppy reasoning. Note however, that there is already evidence of ST398 being rare in healthy colonisation in a European population based on strain-level shotgun metagenomics study, and in a One Health study in the U.S., which we now mention in the section to bring clarity on this issue. We also added a short section on ST398 in the Discussion to highlight the currently limited ability to draw conclusions concerning the relative virulence of ST398.

12. Line 123: 'After ST10 and ST398, the third most common lineage circulating in Punjab is ST58, which is also frequently found in clinical surveillance in Western countries and was commonly isolated from meat products in a recent One Health study conducted in the U.S.' Why did you use 'also' when ST398 is rarely found in clinical studies in the Europe?

We do apologise for sloppy phrasing, 'also' did refer only to ST58, meaning that it is not only common in our Pakistan study, but also frequently found in Western studies. We have edited the text to avoid such misinterpretation.

13. Line 131: inconsistent usage of the term – HIC sometime mentioned high-income countries

"High-income countries" changed to HICs here.

14. Line 134: above cited studies – kindly cite again as it is ambiguous which above study you are talking about

Citations added.

15. Line 136: a very long and difficult to read sentence. Kindly rephrase

We agree and have divided the sentence into two separate sentences.

16. Line 144: check indicates spelling

Thanks for pointing out this, the extra character must have been caused by file format conversion and has been removed.

17. Line 143: Kindly explain the sentence in a better way. Why is difference in ST69, ST131 and ST648 carriage rates in Hospital and community samples indicative of lower frequencies of sustained carriage? (in which population, why?)

This was indeed an unclear statement, we have revised the text to remove the ambiguity.

18. Line 149 : when were the antibiotics administered? Before/during or after the participation?

The antibiotics were either administered at the Manga Mandi clinic during the first visit after the rectal swab was taken, or between the first and second visit. We have amended the description to make this clearer.

19. Line 158 – use 'being' instead of 'as' the most common agents

Altered as suggested.

20. Line 164: what do you mean by generalist?

We use this term in the common sense of microbial ecology literature, referring to the fact that ST10 is frequently observed in many different host animals and is often cited as a generalist.

21. Line 170: use either 'similarly' or 'also'

Removed the 'also'.

22. Line 180: seems like an incomplete sentence: 'We defined 'missing' as a lineage...'

We have rewritten to remove ambiguity.

23. Line 256-258 – 'captures the impact of antibiotic treatment on the overall diversity in healthy carriage'. This statement is not clear as the criteria of the outpatients being referred for the study is not stated. Also, timing of the antibiotic administration is not stated. Especially when outpatients can easily access antibiotics without prescription. Hence we cannot surmise whether this diversity is in 'healthy' carriage.

We stated that the antibiotics were prescribed either during the first visit or between the first and second visit. Furthermore, we wrote as follows in the Discussion: *“Data on medication history was limited to the Manga Mandi hospital prescription and dispensing registry, since obtaining reliable medical history from the patients themselves proved impossible. Some patients could have received antibiotics from elsewhere or bought them over the counter, but we consider it highly unlikely that any relevant quantities of antibiotics would have been acquired from other sources. Nearly all patients were very poor and all care and medications provided at the Manga Mandi hospital were part of their employment benefits, and they were allowed to visit the clinic during working hours. Thus, there was a strong counter-incentive for them to access medications from other sources, which lends support to representativity of the medical records used in our study.”*

We thus argue that the interpretation is sufficiently justified.

24. Line 319 – the sentence is too long and difficult to understand

We agree and have rewritten it for clarity.

25. Line 391 - (v3.0.0-rc,) remove the comma

Removed.

26. Line 385-396 – Methods – inconsistency in referencing with sometimes the reference appearing within a bracket, before a bracket, after a comma in a bracket)

We have fixed this and apologise for the inconsistency, this was caused by us reformatting the citations from the author-year style to superscripts during the submission.

27. No mention of interview questionnaire (survey) for the participants

Participants were indeed interviewed as stated in the methods section and a questionnaire form was filled in by the interviewer. The interviews were conducted in the Punjabi language unless the mother tongue of the participants was something else, in which case Urdu, the lingua franca of Pakistan, was used. Participants were not compensated or reimbursed in any way. We added the information on the lack of compensation to the manuscript

However in this study the only interview data on participants' age and sex is presented so a more detailed description of the interview does not seem to be warranted. The questionnaires did include data on medical history and antibiotic use, but the vast majority of participants were uneducated and could not answer the questions reliably so we decided not to use the data.

28. No mention of the timeline and year of recruitment and processing of samples (except in the metadata in the supplementary table)

Added as suggested

29. No mention of limitation of collection and storage of stool samples from community and their implications on data quality

We added the following paragraph to the discussion section “Another limitation is the storage of the stool samples of the community volunteers and the follow up samples of the outpatients by the participants before they delivered the samples. They were instructed to collect the stool in the morning, store the sample in shade and deliver it as early during the same day as possible. But the samples were in the jars for a few hours in ambient temperatures, which were often around 40°C during the study period. This may have affected the bacterial composition to some degree compared to the primary samples taken by rectal swabs.”

REVIEWERS' COMMENTS

Reviewer #3 (Remarks to the Author):

Okay with the rebuttal and revisions to the paper. No more comments from my end.